# DNA Methylation of *PXDN* Is Associated with Early-Life Adversity in Adult Mental Disorders

**DOI:** 10.3390/biom14080976

**Published:** 2024-08-09

**Authors:** Susanne Edelmann, Jeysri Balaji, Sarah Pasche, Ariane Wiegand, Vanessa Nieratschker

**Affiliations:** 1Department of Psychiatry and Psychotherapy, University Hospital of Tuebingen, Eberhard Karls University of Tuebingen, 72076 Tuebingen, Germany; 2German Center for Mental Health (DZPG), Partner Site Tuebingen, 72076 Tuebingen, Germany; 3Max Planck Fellow Group Precision Psychiatry, Max Planck Institute of Psychiatry, 80804 Munich, Germany; 4Department of Psychiatry and Psychotherapy, LMU University Hospital, LMU Munich, 80336 Munich, Germany

**Keywords:** DNA methylation, Early-life Adversity (ELA), Mental Disorders, Borderline Personality Disorder (BPD), Major Depressive Disorder (MDD), Social Anxiety Disorder (SAD), *PXDN*

## Abstract

Early-life adversity (ELA) is characterized by exposure to traumatic events during early periods of life, particularly involving emotional, sexual and/or physical adversities during childhood. Mental disorders are strongly influenced by environmental and lifestyle-related risk factors including ELA. However, the molecular link between ELA and the risk of an adult mental disorder is still not fully understood. Evidence is emerging that long-lasting changes in the epigenetic processes regulating gene expression, such as DNA methylation, play an important role in the biological mechanisms linking ELA and mental disorders. Based on a recent study, we analyzed the DNA methylation of a specific CpG site within the gene *PXDN*—cg10888111—in blood in the context of ELA across a set of psychiatric disorders, namely Borderline Personality Disorder (BPD), Major Depressive Disorder (MDD) and Social Anxiety Disorder (SAD), and its potential contribution to their pathogenesis. We found significant hypermethylation in mentally ill patients with high levels of ELA compared to patients with low levels of ELA, whereas cg10888111 methylation in healthy control individuals was not affected by ELA. Further investigations revealed that this effect was driven by the MDD cohort. Providing a direct comparison of cg10888111 DNA methylation in blood in the context of ELA across three mental disorders, our results indicate the role of *PXDN* regulation in the response to ELA in the pathogenesis of mental disorders, especially MDD. Further studies will be needed to validate these results and decipher the corresponding biological network that is involved in the transmission of ELA to an adult mental disorder in general.

## 1. Introduction

The complex interaction between genetic risk alleles and environmental stimuli such as adverse events in early life (early-life adversity; ELA) can modulate the response to stressors later in life. Therefore, those interactions play a major role in the resilience to stress and, thus, in the development of psychopathologies such as Borderline Personality Disorder (BPD), Major Depressive Disorder (MDD) and Social Anxiety Disorder (SAD). The underlying molecular mechanisms responsible for these effects are still not fully understood. Recent evidence indicates that the interplay between environmental and genetic factors in the development of psychiatric disorders is partially mediated by epigenetic regulation [1,2]. Environmental stimuli and stress factors shape the epigenome, which in turn is a regulator of gene expression (reviewed in [3]). Epigenetic alterations following ELA might, therefore, potentially lead to changes in gene expression, resulting in the dysregulation of important biological pathways, which is the basis for structural changes as a response to environmental stress.

An association between ELA as a stressor, long-term epigenetic alterations and lasting gene expression changes in peripheral tissue and the brain has been identified in several studies. For example, in baboons, early-life adversity, i.e., resource limitation in their environment, was associated with aberrant DNA methylation patterns in adults [4]. In the human brain, Lutz et al. (2021) reported that ELA was associated with epigenetic, as well as transcriptomic, changes, indicating that immune-related and small GTPase signaling pathways are consistently impaired in the amygdala of ELA individuals [5].

Mental disorders, for which ELA is particularly described as strong risk factor, include MDD, BPD and anxiety disorders, e.g., SAD [6]. SAD is a psychiatric disorder characterized by severe fear of social situations and avoidance of these, and it is especially prevalent in women [7]. MDD is one of the most commonly occurring mental diseases, with a lifetime prevalence of 15.7% in Germany, and it has a major impact on quality of life, autonomy, social integration and life expectancy [8]. BPD is a severe psychiatric disorder associated with substantial impairment in daily life, a high risk of self-mutilation and suicidal tendencies [9]. As mentioned before, ELA may contribute to the risk of those mental disorders by modulating the response to additional stressors on an epigenetic and/or transcriptomic level. For example, Kos et al. (2023) identified cell type-specific changes in the transcriptional states of glutamatergic and GABAergic neurons in the ventral hippocampi of ELA mice after exposure to acute social stress in adulthood [10]. Furthermore, SAD—as a psychiatric disorder corresponding to high levels of social stress—was shown to be associated with ELA in an indirect manner by the aberrant expression of genes involved in neuronal immune signaling pathways [11]. On a gene regulatory level, CpG methylation in *BDNF* and *NR3C1* mediated ~20% of the association between childhood trauma and depressive symptoms [12]. Additionally, several epigenetic modifications such as brain *SLC6A4* methylation or the expression of the miRNA miR-450a-2-3 were associated with the link between ELA and adolescent depression (for a review, see [13]). Furthermore, the differential DNA methylation of, e.g., *FKBP5* or oligodendrocyte genes (e.g., *LINGO3* and *POU3F1*) was identified as a potential molecular link between ELA and adult MDD (for a review, see [14]). In a cross-species and cross-tissue approach, Nieratschker et al. identified the differential methylation of *MORC1* in response to ELA, for which an association with MDD was suggested via gene-set analysis from a genome-wide association study [15]. In the blood of a female BPD cohort, Teschler et al. (2013, 2016) identified increased methylation at the CpG sites of several genes, including *APBA2* and *PRIMA1* [16,17]. In an epigenome-wide association study (EWAS) conducted by Arranz et al. (2021), the authors identified several epigenetic alterations potentially modulating the development of BPD following the exposure of ELA [18]. Interestingly, comparisons between BPD patients with and without ELA identified changes in cg10888111 methylation within the gene *PXDN* that were, although only nominally significant in the EWAS performed in the discovery cohort, significant when validated with a targeted EpiTYPER assay in the same cohort [18]. Additionally, the significantly lower methylation level of *PXDN*, a gene that was associated with intellectual disability and obesity before [19], was also found in a replication cohort when comparing BPD patients with high levels of ELA to patients with low levels of ELA and healthy control individuals [18]. Therefore, this CpG site can be assumed to be relevant for mental disorders such as BPD in the context of ELA.

The aim of this study was, therefore, to contribute to a better understanding of the biological mechanisms involved in the association between ELA and adult mental disorders by investigating whether epigenetic alterations within the gene *PXDN* could be a molecular link between ELA and mental disorders strongly negatively influenced by ELA (especially BPD, MDD and SAD). Therefore, we analyzed the methylation levels of the CpG site cg10888111 identified by Arranz et al. in a large cohort comprising patients suffering from either BPD, MDD or SAD with high or low levels of ELA, as well as healthy control individuals with and without the experience of ELA, respectively.

## 2. Methods

### 2.1. Study Population

In total, 358 participants of European ancestry between 18 and 65 years of age were included in this study. The BPD substudy participants were recruited in 2013–2016, as described in Knoblich et al. (2017) [20]. BPD patients were diagnosed according to the International Personality Disorder Examination (IPDE) and met at least five diagnostic criteria for BPD, as defined in DSM-IV (the fourth version of *The Diagnostic and Statistical Manual of Mental Disorders*, [21]) [20,22,23]. Healthy control participants were matched for age and sex. Participants from the MDD substudy were recruited in 2016–2017, as mentioned in Thomas et al. (2020) [24]. MDD patients were diagnosed according to the DSM-IV criteria by experienced clinicians at the Department of Psychiatry and Psychotherapy, University Hospital Tuebingen, Germany [15,24]. Healthy control individuals were matched for age, sex and ELA. Patients suffering from SAD diagnosed using the Structured Clinical Interview for DSM-IV (SCID) and healthy control individuals (matched for age, sex and ELA) from the SAD substudy were recruited at the University of Tuebingen from 2017 to 2020, as described in Wiegand et al. (2021) [25]. Therefore, the combined overall cohort consisted of 40 BPD patients with 53 control individuals [22], 64 MDD patients with 64 healthy control individuals [24] and 65 individuals suffering from SAD with 72 healthy control individuals (Table 1) [11,25].

ELA was assessed in all three substudies using the Childhood Trauma Questionnaire (CTQ), which measures five dimensions (further referred to as subscales) of maltreatment: emotional and physical neglect and emotional, physical and sexual abuse [26,27]. Responses were measured using a 5-point Likert scale (1 = never true; 2 = rarely true; 3 = sometimes true; 4 = often true; 5 = very often true). Each subscale was represented by five questions with a score range from 5 to 25. Participants with at least a moderate score in one of the five categories (sexual abuse: >8; physical abuse: >10; physical neglect: >10; emotional abuse: >13; emotional neglect: >15) were classified as participants with high levels of ELA [26,27]. Thus, in the initial analysis, four groups emerged: (1) control participants without a mental disorder and low levels of ELA (n = 131), (2) control participants without a mental disorder and high levels of ELA (n = 58), (3) participants suffering from a mental disorder with low levels of ELA (n = 74) and (4) participants suffering from a mental disorder with high levels of ELA (n = 95). Within the separate cohorts, there were 38 BPD patients with high levels of ELA and 2 BPD patients with low levels of ELA, whereas amongst the matched healthy control individuals in this cohort, there were 10 individuals with high levels of ELA and 43 individuals with low levels of ELA. The MDD cohort was composed of 32 MDD patients with high levels of ELA, 32 MDD patients with low levels of ELA and 26 matched healthy control individuals with high and 38 with low levels of ELA. The SAD cohort consisted of 25 individuals with SAD with high and 40 with low levels of ELA; from the matched control individuals, we revealed 22 high and 50 low ELA levels.

All participants gave written informed consent to participate in the experimental procedure prior to inclusion in this study. This study was performed in accordance with the Declaration of Helsinki and approved by the University of Tuebingen local ethics committee.

### 2.2. PXDN DNA Methylation Analysis in Whole Blood

Peripheral venous blood was drawn in Ethylenediaminetetraacetic tubes (EDTA, BD, Heidelberg, Germany) from all participants. Samples were frozen and stored at −80 °C until further analysis. Genomic DNA was isolated from whole blood using the QIAamp DNA Blood maxi Kit (Qiagen, Hilden, Germany) [28,29] following the manufacturer’s protocol. A total of 500 ng of genomic DNA was bisulfite converted using the EpiTect Fast Bisulfite Kit (Qiagen) [30,31] according to the manufacturer’s instructions. Bisulfite converted DNA was eluted in 20 µL elution buffer and stored at −20 °C until further analysis.

Epigenome-wide DNA methylation data adjusted to cell-type ratio estimates were available from the SAD substudy, as specified in Wiegand et al., 2021 [25]. Starting from the CpG previously identified, i.e., cg10888111, this study focused on amplifying a region in the last exon of *PXDN* spanning cg10888111 (hg38, chr:2 1,632,996–1,633,597). Therefore, this CpG site was extracted from the epigenome-wide data using custom Python (version 3.11.5) and MATLAB (version R2023b) scripts.

For the MDD and BPD subcohort, region-specific PCRs were conducted using the PyroMark PCR Kit (Qiagen) [20,22,23] according to manufacturer’s protocol to amplify a region also spanning cg10888111 with the following primers: forward (fwd): 5′-TATATAATTTGAAGTTAGATAGT-3′ and reverse (rev): 5′-Biotin-ATCCCATTATATATCTAATACC-3′. The successful amplification and specificity of the PCR products were verified and visualized via agarose gel electrophoresis. DNA methylation levels were analyzed by pyrosequencing using the PyroMark Q24 system with the sequencing primer (seq): 5′-TTTGGGAAGAGTTA-3′ and the corresponding PyroMark Q24 Software 2.0 (Qiagen) [32,33]. Each sample was amplified twice, and both amplicons were sequenced as technical replicates. The mean percentage was used for further analyses. However, replicates revealing a deviation ≥ 3% were repeated. To detect the disparate amplification of unmethylated DNA fragments, a titration assay using standardized bisulfite-converted control DNA samples (EpiTect Control DNA, Qiagen) [34] with established DNA methylation levels of 0%, 25%, 50%, 75% and 100% was performed.

## 3. Data Analyses

### 3.1. Statistical Analysis

All data analyses were performed using the software environment R (version 4.1.2) and Python (version 3.11.5). Statistical tests, which were available within the R package ggpubr (version 0.6.0) [35] or the Python package scipy.stat (version 1.11.1) [36], were used depending on the analysis specified in the following sections.

To investigate the effects of Disorder, ELA and their interaction on cg10888111 DNA methylation levels, a quantile regression model (using the R package *quantreg* (version 5.98) [37]) was fitted, including age, sex and substudy (BPD, MDD and SAD) as covariates, using the following formula: cg10888111~Disorder*ELA + Age + Sex + Cohort; this defines tau as 0.5 to estimate the median, providing a measure of central tendency that is robust to outliers. In the same way, the quantile regression model was fitted per cohort but without the substudy as covariate.

When the Wilcoxon Mann–Whitney rank sum test was used to compare the *PXDN* methylation levels between (sub)groups, the Benjamini–Hochberg procedure [38] was used to correct for multiple testing and, therefore, protect against false-positive or Type 1 errors. Spearman’s correlation analysis was used to compare continuous variables. In case of multiple testing, the Benjamini–Hochberg correction was performed and an adjusted *p*-value was calculated for the respective number of tests. An adjusted *p*-value (*p*.adj.) < 0.05 was considered to be significant.

### 3.2. Demographic and Clinical Information

The normality of data was tested using the Shapiro–Wilk test (Appendix A). Testing with the Shapiro–Wilk method revealed non-normal distributions for all variables. Therefore, the comparison of the trait medians between the independent groups was performed using the Wilcoxon Mann–Whitney rank sum test.

## 4. Results

### 4.1. Demographic and Clinical Information

The sample characteristics with respect to the four groups emerging from the factors adult mental disorder (further referred to as aMD) and ELA are shown in Table 2.

Between the aMD group and the healthy control group, there was a significant difference in age (n = 357, W = 13,067 and *p*.adj. = 0.005 (n_tests_ = 4), as shown in Appendix A, with a mean age of 32 ± 12 years in the aMD group and 28 ± 9 years in the control group, as shown in Appendix A). Sexes were evenly distributed among the aMD and the control group (Pearson’s Chi-square, χ^2^ = 0.70, *p*.adj. = 0.402 (n_tests_ = 2), with 104 female and 65 male participants in the aMD group and 107 female and 82 male participants in the control group).

The group with low levels of ELA was significantly younger (with a mean age of 28 ± 10 years; Appendix A) than the group with high levels of ELA (n = 358, W = 11,560, *p*.adj. < 0.001 (n_tests_ = 4), as shown in Appendix A, with a mean age of 32 ± 11 years, as shown in Appendix A). Additionally, there were significantly more female than male individuals in the group with low levels of ELA (with 132 female and 73 male participants) compared to an even distribution of sexes in the group with high levels of ELA (Pearson’s Chi-square test, χ^2^ = 5.37 and *p*.adj. = 0.041 (n_tests_ = 2), with 79 female and 74 male participants; Appendix A).

The total score of the CTQ of our overall cohort was significantly different with respect to sex (n = 358, W = 18,279, *p*.adj. = 0.005, n_tests_ = 4; Appendix A), and it correlated positively with age (Spearman’s rho = 0.27, *p*.adj. < 0.001, n_tests_ = 2; Appendix A). However, as the cg10888111 DNAm levels were neither associated with sex (n = 358, W = 16,507, *p*.adj. = 0.300 (n_tests_ = 4), Appendix A) nor correlated with age (Spearman’s rho = 0.08, *p*.adj. = 0.111, n_tests_ = 2, Appendix A), we assumed that these variables did not affect *PXDN* methylation, although age and sex were heterogeneously distributed between the analyzed groups. Participants in the ELA group reported emotional neglect in 93 cases, emotional abuse in 93 cases, physical neglect in 53 cases, physical abuse in 34 cases and sexual abuse in 48 cases.

The distribution of the variables within the four emerging groups when dividing the sample by the combination of aMD and ELA is shown in Table 2. While the patient and the control group (further referred to as HC for healthy controls) were approximately the same size (n_aMD_ = 168, n_HC_ = 189), the number of individuals per combinational subgroup (aMD/high ELA, aMD/low ELA, HC/high ELA and HC/low ELA) varied within the groups. The aMD/high ELA group consisted of more individuals (n = 95) compared to the aMD/low ELA group (n = 73), whereas the HC/high ELA group consisted of 58 individuals compared to 131 HC/low ELA individuals (Table 2). The total CTQ was significantly higher in the groups with high levels of ELA (aMD/high ELA (61.7 ± 18.4) and HC/high ELA (50.9 ± 12.3)) compared to the groups with low levels of ELA (aMD/low ELA (32.8 ± 5.1) and HC/low ELA (33.9 ± 4.8), as shown in Table 1, along with the Wilcoxon test results of single comparisons given in Appendix A). Interestingly, mentally ill patients scored significantly higher in the median total CTQ within the same subgroups compared to healthy control individuals (aMD/high ELA vs. HC/high ELA with a difference of 7.5, n = 168, W = 3759, *p*.adj. < 0.001, aMD/low ELA vs. HC/low ELA with a difference of 1.5, n = 189, W = 6207, *p*.adj < 0.001; Appendix A).

### 4.2. Distribution of PXDN DNA Methylation Levels between Patients with a Mental Disorder with High or Low Levels of ELA and Healthy Control Individuals with High or Low Levels of ELA

To investigate the potential of *PXDN* DNA methylation as a biomarker for ELA and decipher potential molecular regulatory key players for the link between ELA and an adult mental disorder, we analyzed cg10888111, which is located in the last exon and found to be associated with ELA in a BPD cohort created before in [18]. Therefore, as presented above, we combined our three available cohorts (BPD, SAD and MDD patients with respective matching healthy control subjects). We compared the DNA methylation level of cg10888111 between aMD and HC, as well as between the groups of individuals with high and low levels of ELA.

The distribution of cg10888111 methylation revealed a bimodal pattern with peaks around 96.5% and 99% in both comparisons (Figure 1). However, in the context of ELA, the distribution differed between the groups (Figure 1B), which was not visible in the context of aMD (Figure 1A). Therefore, in the group with high levels of ELA without respect to aMD, more individuals revealed a higher cg10888111 methylation of approximately 99.0% compared to individuals with low ELA levels with a methylation level of 96.5% (Figure 1B).

A comparison of the median DNA methylation levels of cg10888111 did not reveal any significant difference between aMD (median ± IQR: 96.6% ± 1.64%) and HC (96.6% ± 1.33%, as shown in Figure 2A; n = 358, W = 16,034, *p* = 0.949). However, individuals with high levels of ELA (96.8% ± 2.12%) were significantly hypermethylated compared to individuals who experienced less or no ELA (96.6% ± 1.07%, n = 358, W = 13,265, *p*.adj. = 0.025; Figure 2B).

When comparing the four groups with high and low levels of ELA, respectively, and accounting for their aMD results, the cg10888111 hypermethylation of individuals who experienced ELA compared to individuals who experienced less or no ELA was only observed within the aMD group: mental disorder patients with high levels of ELA tended to show higher cg10888111 DNA methylation compared to patients with low levels of ELA (aMD/high ELA: 97.0% ± 2.11%; aMD/low ELA: 96.4% ± 0.86%): n = 169, W = 4375, *p*.adj. = 0.039; Figure 3). In contrast, cg10888111 methylation levels did not differ upon ELA status within the HC group (HC/high ELA: 96.6% ± 2.44%, HC/low ELA: 96.7% ± 1.15%, n = 189, W = 3555, *p*.adj. = 0.578; Figure 3). Moreover, there was no significant difference in cg10888111 methylation between any of the other groups.

To control for potential effects of demographic variables, a quantile regression model with the factors disorder, ELA and their interaction and age, sex and substudy as covariates was fitted on the overall cohort. A strong significant effect of the SAD substudy was revealed (*p* < 0.001). However, the effect of ELA on cg10888111 methylation observed in the initial analysis was not significant in this extended model (coefficient = −0.15, *p* = 0.450). Moreover, an interaction between ELA and aMD was not identified by this method (coefficient = 0.04, *p* = 0.892, Appendix A).

### 4.3. PXDN DNA Methylation Correlation with Type of ELA

We then investigated the correlation between cg1088111 methylation and ELA subclasses. The emotional forms of ELA—emotional abuse (Figure 4A) and emotional neglect (Figure 4B)—were significantly, but weakly, positively correlated with cg10888111 methylation levels (emotional abuse: Spearman’s rho = 0.18, *p*.adj. = 0.002; emotional neglect: Spearman’s rho = 0.19, *p* = 0.002) when considering the entire cohort, whereas the physical subtypes of ELA—physical abuse and physical neglect—did not (physical neglect: Spearman’s rho = 0.08, *p*.adj. = 0.114) or had a weak positive correlation (physical abuse: Spearman’s rho = 0.12, *p*.adj. = 0.044) with cg10888111 methylation levels (Appendix A). Sexual abuse—known to associate strongly with the development of BPD [39]—showed a trend towards a weak positive correlation with cg10888111 methylation levels (Spearman’s rho = 0.11, *p*.adj. = 0.051; Appendix A).

### 4.4. PXDN DNA Methylation among the Individual Cohorts

The overall cohort consisted of three different cohorts that were recruited separately within other studies and, therefore, also followed different inclusion and exclusion criteria, as well as recruitment guidelines such as ELA matching. As mentioned in Section 2.1, the BPD cohort included 40 BPD patients, including 2 patients who experienced no or low levels of ELA and 38 patients with high levels of ELA (66.7 ± 22.4; Table 3). A total of 10 of the 53 healthy control individuals experienced high ELA (45.7 ± 7.3), whereas 43 revealed no or low ELA levels (30.7 ± 4.6). More female than male participants took part in this study, with the average age being between 24 and 34 depending on the subgroup (Table 3).

The MDD cohort consisted of 64 MDD patients. A total of 32 of them experienced no or low levels of ELA (32.9 ± 5.3), and 32 experienced high levels of ELA (59.0 ± 15.5; Table 4). Of the 64 HC individuals, 26 experienced high levels of ELA (54.1 ± 8.7) and 38 experienced low levels (31.1 ± 5.6). This cohort was older than the BPD cohort with an average age range of 30–39 (Table 4). Furthermore, more men were recruited than women.

In total, 65 individuals suffering from SAD were recruited. Of those, 25 experienced high levels of ELA (57.5 ± 14.0) and 40 experienced less or no ELA (32.5 ± 5.0). Within the 72 HC individuals, 22 reported high ELA (49.6 ± 10.5) and 50 low or no ELA (29.7 ±4.7, Table 5). The ELA subgroups were balanced for sex and age, with the average age being 24 to 29.

To investigate potential specific links between ELA, *PXDN* DNA methylation and different mental disorders, we analyzed the distribution of cg10888111 methylation among the different cohorts (BPD, MDD and SAD) with respect to ELA (Figure 5). Notably, differential DNA methylation between individuals with high and low levels of ELA was only observed in the MDD cohort, with more individuals with low levels of ELA revealing approximately 97.0% cg10888111 methylation compared to individuals with high levels of ELA, who tended to be hypermethylated at that site with a higher peak at approximately 99.0% (Figure 5C).

Moreover, the cg10888111 methylation levels varied between the cohorts, with significantly lower cg10888111 methylation levels present in the SAD cohort compared to the other two cohorts (SAD vs. BPD: n = 230, W = 10,670, *p*.adj. < 0.001 (n_tests_ = 3); SAD vs. MDD: n = 265, W = 13,619, *p*.adj. < 0.001 (n_tests_ = 3); Table 6).

As we observed an effect of the different cohorts themselves on cg10888111 methylation levels, we then analyzed the impact of the mental disorder in combination with ELA on cg10888111 methylation in each group separately.

There was no difference in cg10888111 methylation levels in the context of mental disorder (BPD, MDD or SAD) in the separate cohorts (BPD: n = 93, W = 1018, *p*.adj. = 0.924; MDD: n = 128, W = 2028, *p*.adj. = 0.924; SAD: n = 137, W = 2205, *p*.adj. = 0.924; Figure 6A–C). Furthermore, the results of the overall cohort concerning ELA-dependent cg10888111 hypermethylation could only be replicated in the MDD cohort.

While a significant effect of ELA was found in the MDD cohort (n = 128, W = 1500, *p*.adj. = 0.034; Figure 7B) due to higher cg10888111 methylation in individuals with high ELA levels, there was no effect of ELA in the BPD (n = 93, W = 1102, *p*.adj. = 0.995; Figure 7A) and SAD (n = 137, W = 2475, *p*.adj. = 0.995; Figure 7B) cohorts.

A quantile regression model including the covariates age and sex was fitted for the MDD cohort. As observed in the overall cohort, the effect of the interaction between ELA and Disorder on cg10888111 methylation in the MDD cohort was not significant in this extended model (Coefficient = −0.62, *p* = 0.524, Appendix A). However, when fitting the model with no interaction between ELA and disorder, a significant effect of ELA on cg10888111 methylation was observed (Coefficient = −1.26, *p* = 0.031, Appendix A).

## 5. Discussion

In this study, we aimed to replicate the differential DNA methylation of *PXDN* in BPD patients and control individuals in the context of ELA, as described in a previous study by Arranz et al. (2021) [18]. In addition, the goal of our study was to analyze the generalizability of the findings and investigate the role of epigenetic dysregulation of *PXDN* as a potential molecular link between ELA and adult mental illness not only in BPD but also other adult mental disorders associated with traumatic childhood experiences, especially MDD and SAD.

We indeed observed the differential DNA methylation of *PXDN*—more specifically cg10888111, which is located in proximity to several other CpG sites in the last exon of the gene—in the context of ELA when analyzing our overall cohort, consisting of BDP, MDD and SAD patients and their respective healthy control individuals. Interestingly, the methylation alteration of cg10888111 between individuals with high and low levels of ELA was especially pronounced when considering the status of mental disorder, with significantly higher *PXDN* methylation detected in mentally ill patients who experienced high levels of ELA compared to mentally ill patients with low levels of ELA, supporting the finding by Arranz et al. (2021) [18], although in our study the effect was reversed. The *PXDN* DNA methylation of healthy control individuals alone was not affected by ELA. Notably, participants suffering from a mental disorder scored significantly higher in the CTQ compared to healthy control individuals, supporting the view that ELA is a risk factor for mental disorders. In healthy control individuals where ELA was less pronounced, it can potentially be compensated for by protective factors, or there might be a dose-dependent relationship between ELA and the risk of mental disorders [40].

When comparing the *PXDN* methylation levels of the group with a diagnosed mental disorder with healthy control individuals disregarding ELA status, no difference in cg10888111 methylation levels was observed, suggesting that differential *PXDN* DNA methylation is not influenced by mental disorders alone but rather by an interaction between ELA and adult psychopathology. Our finding of no significant differences in *PXDN* DNA methylation between healthy control individuals with high and low levels of ELA is also in line with this hypothesis. However, fitting a quantile regression model to account for covariates (i.e., age, sex and substudy) did not reveal a significant effect of ELA, but subcohort status (SAD), on the cg10888111 methylation levels. On one hand, this could be the result of confounding effects of age and sex, although they did not correlate with the DNA methylation values. As we observed significant differences together in age between the subcohorts and between aMD and HC, as well as the subgroups with high and low levels of ELA, we cannot exclude an effect of age on the DNA methylation of *PXDN*, especially as age was generally shown to be affected by DNA methylation (for a review, see [41]) On the other hand, together with the rather small effect observed in this study, the small sample size might have led to a lack of power to conduct this analysis.

Furthermore, we observed differential overall *PXDN* methylation levels between the substudies. As cg10888111 methylation was measured via pyrosequencing in the BPD and the MDD cohorts (displaying similar overall DNA methylation levels) and via the Infinium MethylationEPIC BeadChip (Illumina, San Diego, CA, USA) in the SAD cohort, using two different techniques most likely contributes to imbalances between the cohorts. Moreover, accounting for the individual blood cell type proportions was only possible in the SAD cohort using the Houseman reference data for epigenome-wide approaches [42]. Unfortunately, the blood cell counts of the participants in the MDD and BPD substudies were not available.

However, the observed effect was mainly driven by the cohort of MDD patients and their matched healthy control individuals. Therefore, we fitted a quantile regression model with an interaction term of ELA and disorder on *PXDN* methylation for this substudy, as well as for the overall cohort, which revealed no significant effect. Additionally, we fitted the model for the effect of ELA and disorder on cg10888111 methylation separately, which revealed a significant effect of ELA on the DNA methylation of *PXDN*. Therefore, we cannot mathematically detect an effect of interaction between MDD and ELA on *PXDN* methylation in our subcohort. Nevertheless, an association between ELA and MDD was shown previously (e.g., [43]; for a review, see [44]), and an indirect link between ELA and MDD that is partially regulated on an epigenetic level is still possible, as demonstrated in SAD in a previous work [11].

A possible explanation for the effect of ELA on cg10888111 methylation, especially pronounced in the MDD cohort, may be an age effect, as participants with high levels of ELA were generally older, and the participants in our MDD cohort were older than those in the BPD and SAD cohorts. However, age was not significantly associated with *PXDN* DNA methylation. Nevertheless, we hypothesize that age is a relevant factor for experiencing of ELA, which could have two possible reasons: 1. European Baby Boomers (born 1946–1964) and Generation X (born 1965–1980) grew up during times of significant social upheaval, economic instability and less stringent child protection laws. These factors contributed to higher instances of various forms of childhood trauma, including physical and emotional abuse and neglect. The increasing awareness and better implementation of child protection laws and mental health services in later years have helped to reduce such adversities for Millennials (born 1981–1996) and Generation Z (born 1997–2012) individuals. Additionally, the types of ELA changed between those generations and experiences that are more stressful for younger generations, such as digital bullying, are not well covered by the CTQ. 2. Increasing awareness of individuals and society over time, as well as a greater possibility of patients being treated, may enhance the understanding and awareness of ELA, and it could also be a contributing factor. Therefore, ELA and its perception may be more pronounced in older individuals. The usage of the CTQ itself reveals other limitations, although various studies confirm its consistency and reliability [45,46,47,48], e.g., a certain risk of retrospective bias, especially within patient groups [49,50]; the limited scope of other forms of adversity, such as childhood interpersonal trauma [51], are not captured; and the lack of context in terms of details of the frequency, duration or severity of adverse experiences [52].

Moreover, Arranz et al. (2021) [18] observed hypomethylation upon ELA in their BPD patient cohort. We were not able to replicate these results in our study. First of all, in our overall cohort, *PXDN* methylation levels were increased in individuals who experienced early-life trauma, which is in contrast to their results. As mentioned before, the MDD cohort was found to drive this reversed effect. MDD and BPD are two mental disorders that share factors such as affective instability [53]. MDD often presents as comorbidity in BPD patients [54]. However, these mental disorders also reveal distinct sets of symptoms that may involve distinct epigenetic patterns. Perroud et al. (2011) reported higher overall *NR3C1* exon 1F methylation levels in BPD than in MDD subjects in peripheral blood leucocytes that also correlated with the forms of ELA mentioned in our study [55]. Moreover, whereas Perroud et al. (2013) identified significantly higher methylation levels of *BDNF* in BPD subjects [56], MDD was associated with lower [57] and higher *BDNF* methylation status [58,59]. These results imply that although MDD and BPD share a lot of neurobiological aberrations from healthy control individuals, at the single-gene level, epigenetic differences may be possible.

Additionally, our BPD patients did not reveal *PXDN* methylation upon ELA. The BPD cohort was not recruited with ELA in mind; therefore, there is no balance of individuals with and without ELA between the groups. In particular, in the BPD cohort, this led to a very small number of patients with low levels of ELA (n = 2), which prevented us from accurately replicating this study. However, these numbers support the fact that ELA is associated with the development of mental disorders in adult life.

PXDN is a heme-containing peroxidase best known for its role in external matrix formation, especially the formation of a sulfilimine bond, which cross-links collagen IV in basement membranes via the catalyzed oxidation of bromide to hypobromous acid, providing structure and mechanical stability throughout tissue development, homeostasis and wound healing [60]. It is associated with a variety of human diseases such as obesity and intellectual disability [19], several forms of cancer ([61,62,63]; for a review, see [60]), Autism Spectrum Disorder [64] and, interestingly, Post-Traumatic Stress Disorder (PTSD, [65]). Recent studies have investigated its roles in innate immunity, cardiovascular physiology and diseases and extracellular matrix formation and found that PXDN-generated reactive oxidants are important components for host defense, collagen IV synthesis in basement membrane development and tissue genesis and signaling pathways and homeostasis under physiological conditions (for a review, see [66]).

Hence, its immune-related function is especially interesting in the context of ELA as it is known that stress and trauma impact the immune response [67]. We identified increased CpG methylation in the last exon of the gene. DNA methylation in gene bodies is thought to be positively correlated with gene expression [68], transcriptional elongation and alternative splicing [69]. Gene body methylation could assist in the silencing of potentially detrimental repetitive DNA elements such as LINE1 and *Alu* [70]. Exonic DNA methylation was shown to cause C → T transition mutations, leading to disease-causing mutations in the germline and cancer-causing mutations in somatic cells [71]. However, most evidence points towards the relevance of exon DNA methylation in alternative splicing. Exons were found to be more highly methylated than introns, and transitions in the degree of methylation occur at exon–intron boundaries, possibly suggesting a role for differential DNA methylation in transcript splicing [72]. Manipulating DNA methylation in vivo in a site-specific manner using the deactivated endonuclease Cas9 fused to DNA methylation associated enzymes, Shayevitch et al. (2018) demonstrated that changes in the DNA methylation pattern of alternatively spliced exons, but not constitutively spliced exons or introns, altered inclusion levels [73]. Moreover, induced inhibition of DNA methylation led to alternative splicing events in human cell cultures [74,75]. Furthermore, a positive correlation between methylation density and the exon expression level of intragenic exons was observed [76]. It could, therefore, be assumed that the methylation of the analyzed CpG site could either be involved in alternative splicing of *PXDN* or lead to higher *PXDN* gene expression. Therefore, epigenetic dysregulation, leading to altered expression regulation, in turn leading to an increased expression as an immune-related response to early adversity, may be possible. This hypothesis is supported by the protein PXDN interaction network (Appendix A, [77,78]): Interactions with proteins such as MAPK3 and STAT3, whose gene expression levels were previously associated with the molecular link of ELA and adult mental disorders in [11], suggest the involvement of *PXDN* in neuronal inflammatory signaling as a response to ELA, which may ultimately result in altered neuronal plasticity and brain activity [79,80].

Some mental disorders are correlated with certain subtypes of ELA such as BPD and SAD, in particular significantly correlating with emotional ELA [11,81,82,83]. We investigated the link between cg10888111 methylation levels and identified a significant correlation with emotional neglect and emotional abuse. We also observed a weak but significant correlation of cg10888111 methylation levels and sexual abuse, which was mainly seen in the SAD cohort. A link between *PXDN* and emotional ELA has not been explicitly analyzed previously, but we assume that *PXDN* is involved in a regulatory network that responds to the experience of emotional ELA as part of the inflammatory system, as is known for ELA in general (e.g., [84,85,86]). However, an increase in inflammatory activity was associated with the occurrence of early-life sexual abuse [87]. Several inflammatory markers such as IL-6 and TNF-α—both interaction partners of PXDN—showed up-regulation upon childhood sexual abuse [87], which supports the aberrant regulation of *PXDN* through these forms of childhood adversity.

## 6. Conclusions

The present study is the first comparative analysis of DNA methylation of *PXDN* (cg10888111) in whole blood of a combined and separately analyzed transdiagnostic approach (BPD, MDD and SAD) and their balanced healthy control groups with or without the experience of ELA. Interestingly, the hypermethylation of cg10888111 in the last exon of this gene was identified upon high levels of ELA compared to low levels of ELA in the overall cohort and the MDD cohort, whereas the site was not differentially methylated in patients compared to healthy control individuals. However, this effect did not survive when accounting for covariates such as sex and age in a quantile regression model, although we might have lacked the power to conduct this analysis due to our rather small sample size.

Although potentially driven by the MDD cohort and not significant in an interaction model, patients particularly showed differential *PXDN* methylation between high and low ELA experience, as also previously reported by Arranz et al. (2021) [18] in BPD, even though, in that case, the effect was in the opposite direction, with a hypomethylation in patients with ELA. However, based on previous findings, we hypothesize a contribution of the immune system’s response to ELA to an adult mental disorder (i.e., MDD) via *PXDN* regulation. Further studies with larger sample sizes are needed to shed more light on the possible involvement of the immune system in the transmission of ELA to an adult mental disorder.

## Figures and Tables

**Figure 1 biomolecules-14-00976-f001:**
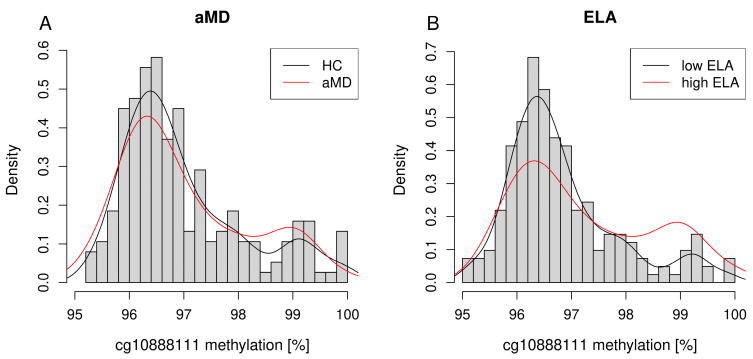
Histograms depicting the cg10888111 methylation levels between (**A**) aMD and HC and (**B**) high ELA and low ELA in the overall cohort.

**Figure 2 biomolecules-14-00976-f002:**
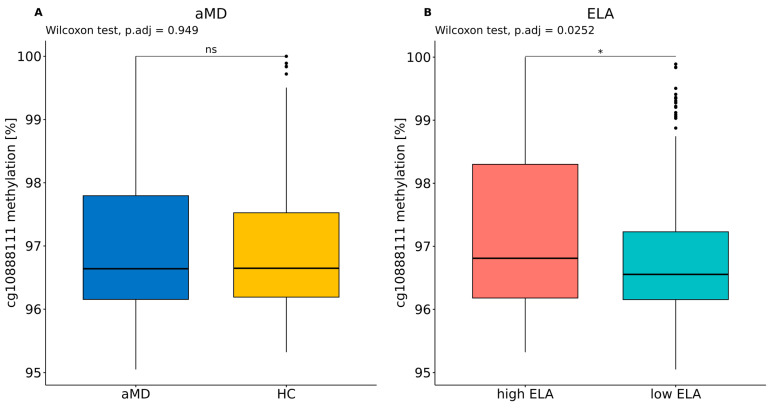
Cg10888111 methylation levels in the context of (**A**) aMD and (**B**) ELA. Wilcoxon rank sum tests and the Benjamini–Hochberg correction were applied (n_tests_ = 2). * *p*.adj. < 0.05. ns: not significant.

**Figure 3 biomolecules-14-00976-f003:**
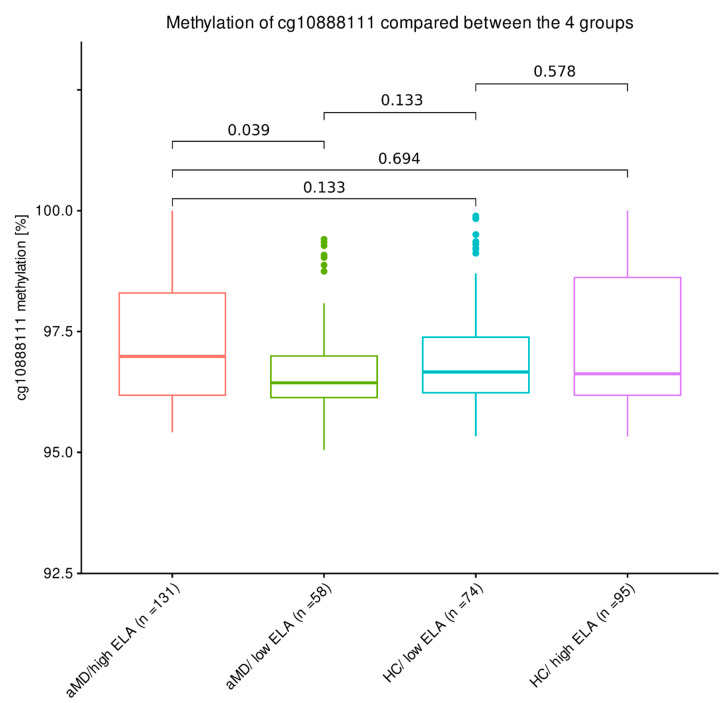
The cg10888111 methylation patterns with respect to aMD and ELA. A Wilcoxon rank sum test was applied in a pairwise comparison. The Benjamini–Hochberg correction was applied (n_tests_ = 5).

**Figure 4 biomolecules-14-00976-f004:**
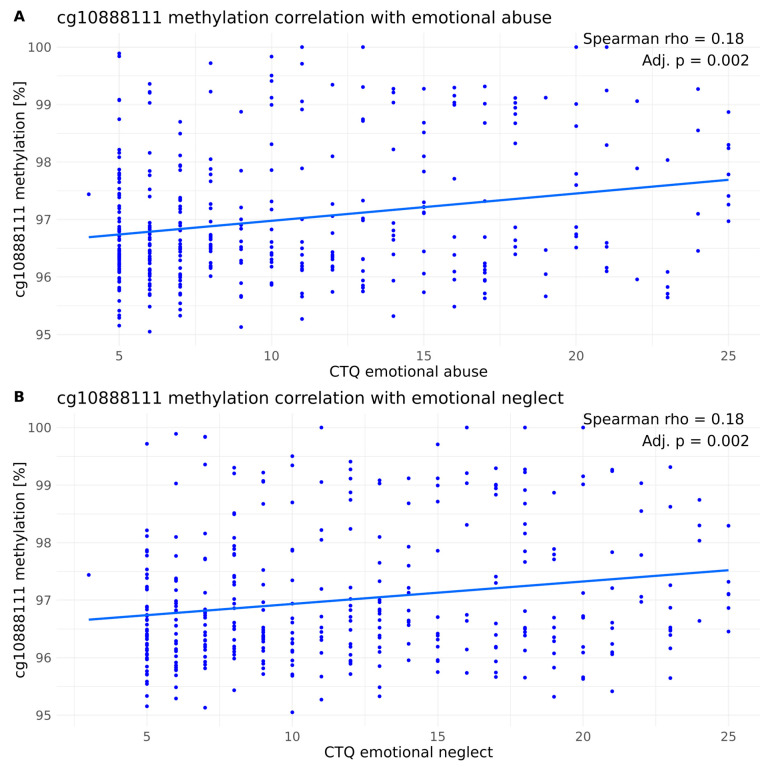
Spearman’ correlation of cg10888111 methylation levels within the overall cohort with (**A**) emotional abuse and (**B**) emotional neglect. The Benjamini–Hochberg correction was applied (n_tests_ = 5).

**Figure 5 biomolecules-14-00976-f005:**
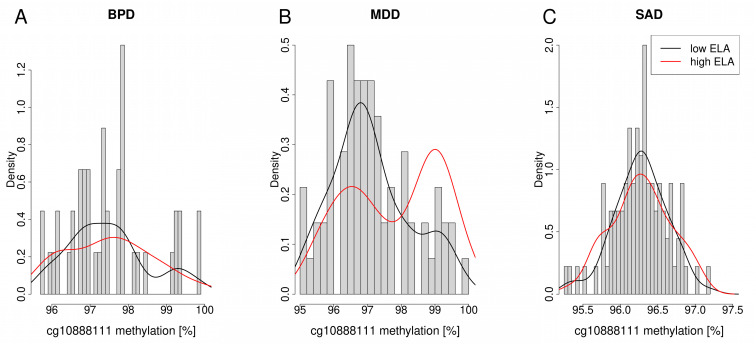
Histograms depicting the cg10888111 methylation levels between high ELA and low ELA in the (**A**) BPD, (**B**) MDD and (**C**) SAD cohorts.

**Figure 6 biomolecules-14-00976-f006:**
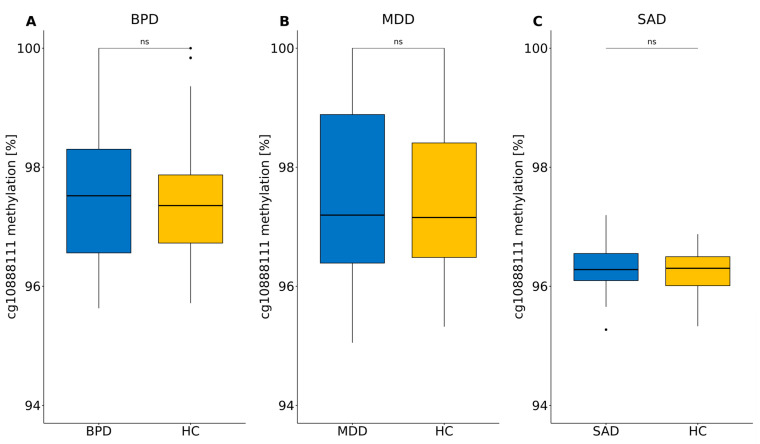
Cg10888111 methylation levels in the context of the respective disorder vs. healthy control individuals in the (**A**) BPD, (**B**) MDD and (**C**) SAD cohorts. Wilcoxon rank sum tests and the Benjamini–Hochberg correction were applied (n_tests_ = 3). ns: not significant.

**Figure 7 biomolecules-14-00976-f007:**
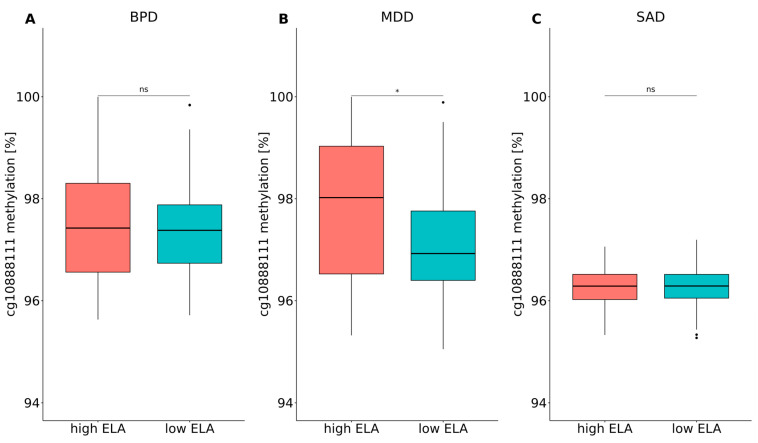
Cg10888111 methylation levels in the context of ELA in the (**A**) BPD, (**B**) MDD and (**C**) SAD cohorts. Wilcoxon rank sum tests and the Benjamini–Hochberg correction were applied (n_tests_ = 3). * *p*.adj. < 0.05. ns: not significant.

**Table 1 biomolecules-14-00976-t001:** Participant numbers for each substudy. HC = healthy control individuals.

	BPD (n = 93)		MDD (n = 128)		SAD (n = 137)	
	Patients	HC	Patients	HC	Patients	HC
*n*	40	53	64	64	65	72

**Table 2 biomolecules-14-00976-t002:** Sample characteristics for the four groups emerging for the factors mental disorder and ELA.

	Mental Disorder (BPD, MDD and SAD)		Healthy Control Individuals	
	High ELA	Low ELA	High ELA	Low ELA
*n*	95	73	58	131
CTQ total score	61.7 ± 18.4	32.8 ± 5.1	50.9 ± 12.3	33.9 ± 4.8
Sex	♀ 55, ♂ 40	♀ 49, ♂ 25	♀ 24, ♂ 34	♀ 83, ♂ 48
Age [y]	33 ± 11	30 ± 12	30 ± 10	28 ± 9

**Table 3 biomolecules-14-00976-t003:** Sample characteristics for the four groups emerging within the BPD cohort in the context of ELA. *NA*: Not available.

	BPD Patients		Healthy Control Individuals	
	High ELA	Low ELA	High ELA	Low ELA
*n*	38	2	10	43
CTQ total score	66.7 ± 22.4	*NA*	45.7 ± 7.3	30.7 ± 4.6
Sex	♀ 32, ♂ 6	♀ 1, ♂ 1	♀ 6, ♂ 4	♀ 40, ♂ 3
Age [y]	32 ± 9	24 ± 3	34 ± 9	27 ± 9

**Table 4 biomolecules-14-00976-t004:** Sample characteristics for the four groups emerging within the MDD cohort in the context of ELA.

	MDD Patients		Healthy Control Individuals	
	High ELA	Low ELA	High ELA	Low ELA
*n*	32	32	26	38
CTQ total score	59.0 ± 15.5	32.9 ± 5.3	54.1 ± 8.7	31.1 ± 5.6
Sex	♀ 7, ♂ 25	♀ 19, ♂ 13	♀ 4, ♂ 22	♀ 13, ♂ 25
Age [y]	39 ± 14	38 ± 14	32 ± 11	30 ± 13

**Table 5 biomolecules-14-00976-t005:** Sample characteristics for the four groups emerging within the SAD cohort in the context of ELA.

	Individuals Suffering from SAD		Healthy Control Individuals	
	High ELA	Low ELA	High ELA	Low ELA
*n*	25	40	22	50
CTQ total score	57.5 ± 14.0	32.5 ± 5.0	49.6 ± 10.5	29.7 ±4.7
Sex	♀ 16, ♂ 9	♀ 29, ♂ 11	♀ 14, ♂ 8	♀ 30, ♂ 20
Age [y]	29 ± 9	24 ± 5	28 ± 9	25 ± 4

**Table 6 biomolecules-14-00976-t006:** Mean and median cg10888111 methylation levels of the BPD, MDD and SAD cohorts. SD = standard deviation; IQR = interquartile range.

Cg10888111 Methylation	BPD	MDD	SAD
Mean [%] ± SD	97.5 ± 1.1	97.4 ± 1.3	96.3 ± 0.4
Median [%] ± IQR	97.4 ± 1.5	97.2 ± 2.3	96.3 ± 0.5

## Data Availability

The raw data supporting the conclusions of this article are available in the Appendix A.

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
