# Peer review of "DNA Methylation of PXDN Is Associated with Early-Life Adversity in Adult Mental Disorders"

_biomolecules, 2024, doi:10.3390/biom14080976_

Round 1

Reviewer 1 Report

Comments and Suggestions for Authors

 The manuscript is very clearly written, and the descriptions of results are nicely presented. The main challenge here is that the primary result goes in the opposite direction from a prior study of the same CpG, and therefore analyses need to do their best to explain why.

Comment 1.

It is somewhat unclear from the manuscript text, but it does appear that the BPD, MDD and SAD individuals (and controls for each) were from different original studies and with different recruitment strategies. If so, the original studies should be named, and their designs explained briefly either in Methods or in a Supplemental document.  Even if not, Methods still requires more information on how these different sets of individuals were assembled and on the recruitment strategies.

It is clear from the data and results shown that these three subsets (BPD, MDD, SAD) are quite different – sex and age distributions, type of ELA, etc. The assumption that age and sex do not affect methylation levels at cg10888111 is a strong one, given that age and sex are well known – generally speaking – to influence methylation levels.

Nevertheless, the authors should consider performing adjusted analyses, adjusting for substudy, age and sex. The Wilcoxon is a nonparametric test, which is a nice choice given that the data are not normally distributed. A similar goal, i.e. robust analyses, but with additional covariates can be achieved by using quantile regression such as in the R package quantreg. Using an “R” style  notation, here is an example:

·       rq(methylation ~ age + sex + ELA + study)

I also recommend transforming the methylation values with a logit transformation to improve their distribution prior to analyzing them.

Comment 2.

Cell type composition has an important effect on methylation levels, but this factor is not mentioned anywhere. If genome-wide methylation levels are available in the SAD cohort, cell type proportions could be estimated using the method of Houseman (see Houseman et al. DNA methylation arrays as surrogate measures of cell mixture distribution. BMC Bioinf. 2012;13:86, or more recent methods such as those compared in https://www.ncbi.nlm.nih.gov/pmc/articles/PMC5307731/.) Unfortunately, this will not be possible for the other 2 sets of study participants unless measurements of cell type proportions have been performed independently by other methods. This must be acknowledged as a limitation.

Comment 3.

Overall the manuscript includes quite a number of tests comparing methylation levels between ELA low and high, or types of ELA, subgroups of participants, correlations, etc. Therefore, using p<0.05 is too liberal for the significance threshold in order to control type 1 error. Please state clearly the number of tests performed. Perhaps some are primary hypotheses and others are secondary hypothesis, in which case give both numbers.  Then discuss your strategy (strategies) for controlling type 1 error. At one point you mention the Benjamini-Hochberg method for controlling false discovery rates, but Methods has no justification of nor discussion of this choice. This is really quite important given the conclusions show an effect in the unanticipated direction.  An expanded discussion addressing this lack of replication of the direction of effect, with all caveats, would be helpful.

Comment 4.

Table 1 should also provide numbers of BPD, SAD, MDD, and numbers in each corresponding control group, for each substudy.

Comment 5.

Please show Figure 1 as histograms rather than smoothed density plots, particularly since the sample sizes are relatively small.

Reviewer 2 Report

Comments and Suggestions for Authors

MAIN 

This in an interesting work that starting from a previous investigation of the same group wich reported an epigenome wide DNA methylation analysis, explored the DNA methylation of a specific CpG site within the gene PXDN, a gene associated with intellectual disability and obesity, in blood in the context of early life adversity (stressful experiences that a person faces during their childhood or early developmental years) across various psychiatric disorders. As main findings, significant hypermethylation in mentally ill patients with high levels of early life adversity compared to patients with low levels of early life adversity, whereas cg10888111 methylation of healthy control individuals was not affected by ELA.

This interesting, well written, and well organized work makes a valuable contribution to the field of early life adversity and particularly the implication of epigenetics in the development of these traumatic events. The interaction between environmental stimuli such as adverse events in early life and epigenetics is an interesting, yet limitingly explored research area. I believe that the experimental design is strong and data presented are interesting and well described. Conclusions are well supported by the epigenetic findings on PXDN.

Considering the topic covered, the adequate scientific level of the manuscript as well as the detailed figures, I consider this nice work suitable for biomolecules. Below are some comments and suggestions for improving the manuscript.

MAJOR

- As stated by the authors, cg10888111 is located within the last exon of the PXDN gene. Therefore, it is unlikely that this CpG site has a biological function, such as localization within a gene promoter region, which is predominantly located upstream of the transcription start site of a gene. How can this epigenetic modification be biologically relevant in terms of gene expression regulation and influence a phenotype? The authors are encouraged to address this aspect in the discussion.

MINOR 

1. Regarding the abstract, I recommend reviewing the journal’s guidelines on whether supporting references should be included. Typically, journals discourage including such quotation in the abstract. In case, “Arranz et al. (2021)” should be removed and use forms such as “Based on a recent study, we analysed…..”

2. Please include more supporting references in the methods, especially the section 2.2. For instance, for the EpiTect Fast Bisulfite Kit (Qiagen), please include these two references which reported the usage of this kit PMID: 27223861 and PMID: 26247357

3. For a better reading, I suggest moving the sentences on the Epigenome wide DNA methylation data at the beginning of the section 2.2 and state that “…starting from the CpG previously identified, i.e., cg10888111, the study focused on amplify a region in the last exon of PXDN spanning cg10888111 (hg38, chr:2 1,632,996-1,633,597) with the following…..”

4. Results, I discourage the mentioning of every statistical test employed, I believe that it quite redundant with the methods. They can be removed for an improved reading o the text 

MAIN 

This in an interesting work that starting from a previous investigation of the same group wich reported an epigenome wide DNA methylation analysis, explored the DNA methylation of a specific CpG site within the gene PXDN, a gene associated with intellectual disability and obesity, in blood in the context of early life adversity (stressful experiences that a person faces during their childhood or early developmental years) across various psychiatric disorders. As main findings, significant hypermethylation in mentally ill patients with high levels of early life adversity compared to patients with low levels of early life adversity, whereas cg10888111 methylation of healthy control individuals was not affected by ELA.

This interesting, well written, and well organized work makes a valuable contribution to the field of early life adversity and particularly the implication of epigenetics in the development of these traumatic events. The interaction between environmental stimuli such as adverse events in early life and epigenetics is an interesting, yet limitingly explored research area. I believe that the experimental design is strong and data presented are interesting and well described. Conclusions are well supported by the epigenetic findings on PXDN.

Considering the topic covered, the adequate scientific level of the manuscript as well as the detailed figures, I consider this nice work suitable for biomolecules. Below are some comments and suggestions for improving the manuscript.

MAJOR

- As stated by the authors, cg10888111 is located within the last exon of the PXDN gene. Therefore, it is unlikely that this CpG site has a biological function, such as localization within a gene promoter region, which is predominantly located upstream of the transcription start site of a gene. How can this epigenetic modification be biologically relevant in terms of gene expression regulation and influence a phenotype? The authors are encouraged to address this aspect in the discussion.

MINOR 

1. Regarding the abstract, I recommend reviewing the journal’s guidelines on whether supporting references should be included. Typically, journals discourage including such quotation in the abstract. In case, “Arranz et al. (2021)” should be removed and use forms such as “Based on a recent study, we analysed…..”

2. Please include more supporting references in the methods, especially the section 2.2. For instance, for the EpiTect Fast Bisulfite Kit (Qiagen), please include these two references which reported the usage of this kit PMID: 27223861 and PMID: 26247357

3. For a better reading, I suggest moving the sentences on the Epigenome wide DNA methylation data at the beginning of the section 2.2 and state that “…starting from the CpG previously identified, i.e., cg10888111, the study focused on amplify a region in the last exon of PXDN spanning cg10888111 (hg38, chr:2 1,632,996-1,633,597) with the following…..”

4. Results, I discourage the mentioning of every statistical test employed, I believe that it quite redundant with the methods. They can be removed for an improved reading o the text 

Round 2

Reviewer 1 Report

Comments and Suggestions for Authors

The manuscript has been appropriately revised and results are presented with nuanced interpretation.